# A Study on the Effects of Digital Finance on Green Low-Carbon Circular Development Based on Machine Learning Models

**Xuewei Zhang [1,2], Xiaoqing Ai [3,*], Xiaoxiang Wang [3,*], Gang Zong [3] and Jinghao Zhang [1]**

[1]  School of Economics and Management, Inner Mongolia University, Hohhot 010021, China; zhangxuewei219@126.com (X.Z.); 15147322831@163.com (J.Z.)

[2]  Institute of Geographic Sciences and Natural Resources Research, CAS, Beijing 100101, China

[3]  School of Economics and Management, Beijing University of Technology, Beijing 100124, China; 15847206038@163.com

[*]  Correspondence: wangxiaoxiang626@163.com (X.W.); axq@bjut.edu.cn (X.A.)

**Abstract:** With technological transformations such as big data, blockchain, artificial intelligence, and cloud computing, digital techniques are infiltrating the field of finance. Digital finance (DF) is a resource-saving and environmentally friendly innovative financial service. It shows great green attributes and can drive the flow of financial resources towards environmentally-friendly enterprises, thereby promoting green low-carbon circular development (GLCD). However, few studies have explored the coupling mechanism between DF and GLCD. To fill this gap, this paper explores the effect of DF on GLCD, and established a mediating effect model to investigate the mechanism of DF in promoting GLCD. Additionally, this paper established a random forest model and a CatBoost model based on machine learning to examine the relative importance of DF and the factors affecting GLCD. The results show that DF has significant positive effects on GLCD, and technological innovation plays a key role in the effect of DF on GLCD; meanwhile, the effect of DF on GLCD shows nonlinear features with an increasing "marginal effect"; moreover, both DF and conventional factors have significant impacts on GLCD. Our study highlights the effect of DF on GLCD and underscores the importance of developing policies for DF and GLCD. This study provides an empirical basis and path reference for DF to achieve "carbon peak, carbon neutralization" in China.

**Keywords:** DF; GLCD; machine learning; double-carbon target

**MSC:** 91B62; 91B76

## 1. Introduction

Over the past decades, China has made great achievements in economic development, and has become the largest energy consumer globally. However, there is an increasingly severe contradiction between economic growth and the ecological environment [1]. Hence, the Chinese government proposed a green and low-carbon development concept aiming to achieve harmony between human beings and nature, thereby promoting coordinated development of economic growth and environmental protection. Currently, green low-carbon circular development (GLCD) has become a global trend due to its significant role in ecological civilization and the development of a beautiful China [2]. The key areas of green low-carbon industries in China are relatively underdeveloped, and future development requires great financial support. However, the conventional financial mode is constrained by structural mismatch and fails to guide a green and low-carbon transformation. With the continuous integration of Emerging technologies such as big data, blockchain, artificial intelligence, and cloud computing with traditional finance, digital finance (DF) came into being [3], which can alleviate the mismatch of financial resources [4,5]. Specifically, DF can allocate capital towards green and low-carbon projects and accelerate the research and application of advanced green and low-carbon technologies [6], thereby facilitating

the ecological environment and green development [7–9]. In the context of energy saving and carbon emission reduction [10], green and low-carbon development is an inevitable requirement for China's sustainable economic development [11–14].

Previous studies have investigated the effect of DF on economic growth, industrial structure, export, household consumption, and income inequality [15–19]. Early studies focused on the impacts of financial development on environmental protection, but no conclusion is drawn [20–23]. Recently, the effect of DF on GLCD has attracted increasing attention. Most scholars believe that DF has positive impacts on GLCD. For example, Wu et al. claimed that the digitization level in DF has a positive impact on carbon emission efficiency [24]; Razzaq and Yang found that DF can facilitate green growth by supporting the digital transformation of enterprises and addressing energy deficit [25]. From the perspective of distorted element configuration, Yang conducted an empirical test regarding the role and working mechanism of DF in carbon emission reduction based on the panel data of 280 cities in China from 2011 to 2019. The results indicated that DF facilitates rational resource allocation and reduction in carbon emissions [26]. From the perspective of capital bias towards technological advances, Li analyzed the impacts of DF on low-carbon energy transformation based on the panel data of 283 cities in China, and it was found that DF can improve carbon total factor productivity and the proportion of clean energy in total energy consumption, exhibiting positive impacts on low-carbon energy transformation [27].

Some scholars have found that DF has nonlinear impacts on green growth [28]: it has negligible impacts on green growth in the early stages, but the impact continuously increases with the level of DF [29]. Nevertheless, Fang et al. (2020) pointed out that financial development stimulates economic growth in China at the cost of increased carbon emissions [30]. By using autoregressive distribution lag-bound test methods, some studies revealed the positive impacts of financial development on carbon emission [31,32].

However, there are some gaps in the literature that limit our understanding of how DF contribute to GLCD. First, although research on low-carbon circular development has attracted academic interest, few studies have considered the multidimensional nature of the GLCD. And the impact mechanism of DF on low-carbon circular development is unknown and fails to provide policy guidance for carbon emission control and environmental improvement in China. The current study aims to clarify the effect of DF on low-carbon circular development in China, then hypothesized and validated the impact mechanism of DF on GLCD. As it pertains to offering both theoretical insights and practical implications, this research contributes to the field in three critical ways. First, the GLCD indicator system is established from the perspectives of ecological civilization and green economy and measured by adopting the spatiotemporal range entropy weight method. Second, the impact mechanism of DF on GLCD is clarified, and the effect of DF on GLCD through technological innovation are clarified by using the mediating effect model. Third, the nonlinear features and regional heterogeneity of the impact of DF on GLCD are investigated to clarify the mechanism of DF affecting GLCD. To sum up, this study aims to provide references for the positive impacts of DF on GLCD by conducting empirical research from the perspectives of the ecological economy and digital economy.

The rest of this paper is as follows: Section 2 investigates the working mechanism of DF affecting GLCD, as well as the corresponding hypotheses; Section 3 proposes a GLCD indicator system and its measurement; Section 4 constructs econometrics models, mediating effect models, and threshold effect models, and introduces relevant variables and data sources; Section 5 presents an empirical test of the impacts of DF on GLCD in terms of extent, mechanism, and threshold effect; Section 6 presents an analysis of the significance of different variables by using machine learning models, heterogenic analysis by regions and basins, and robustness testing of reliability of the results, and provides some recommendations.

## 2. Theoretical Analysis and Hypotheses

### 2.1. Effects of DF on GLCD

Due to limitations in cost and technology, conventional finance cannot provide sufficient support for GLCD in China [33,34]. With advances in the Internet, big data, and artificial intelligence, DF has emerged as an integration of conventional finance and digital techniques. As a novel financial pattern, DF has broken the bottleneck of conventional finance by increasing coverage and reducing transaction costs [35]. DF can reduce the financing threshold with its inclusiveness and assists financial institutions such as banks in risk assessment of real-time transactions and behavioral characteristics of small and medium-sized enterprises by using machine learning, thereby alleviating information asymmetry issues in lending [36]. Meanwhile, the unique and superior information collection and processing capabilities of DF allow optimizing the marketization configuration of capital factors. This enhances the interaction efficiency of information data in the factor market and the matching efficiency of different subjects [37,38] and improves the utilization efficiency of capital factors, thereby providing efficient and convenient financial services to various economic entities. Moreover, DF is essentially a green development mode, and it can accurately identify green and low-carbon features of enterprises, drive the capital to environmentally-friendly enterprises [39], optimize the allocation of credit funds among environmental protection industries, and facilitate the transformation of energy-intensive industries to high profit, high value-added, and low-carbon industries [40,41], thereby promoting the development of the green low-carbon circular economy.

**Hypothesis 1 (H1):** *DF can improve utilization efficiency of capital factor, and promote GLCD.*

### 2.2. Effects of DF on GLCD through Technological Innovation

DF can alleviate the financing constraints of enterprises, and promote innovation and upgrading of the financial industry in various aspects [42]. It increases the coverage of financial services and the depth of financial services, thereby alleviating the mismatch of financial resources [43,44]. Therefore, DF can accelerate financial digitization and provide sufficient financial resources for small and micro enterprises and private enterprises with market potential [45,46]. Financing incentives can strengthen the drive of enterprises on innovations. Digital techniques such as big data and cloud computing allow searching, analysis, and decision-making of consumption information of different users, thereby providing financing support for innovation activities with a high probability of success, which helps entities to explore innovations and improves the output and transformation of technological research and development achievements [47]. In this way, a comprehensive graph of technology and financial application scenarios is developed, and the financing channels for innovations are expanded, thereby promoting urban innovations.

With positive impacts on economic growth, carbon emission, and environmental protection, technological innovation is a key driving factor of GLCD [48,49]. It allows precise labor division, collaboration, and production in the production process, which effectively reduces resource factor loss during production and transmission and enhances energy utilization efficiency. In this way, technological innovation can achieve carbon-free, carbon reduction, and decarbonization before, during, and after the production process, thus promoting intensification and circular utilization of resources. Meanwhile, the technological innovation can optimize and upgrade carbon treatment and pollution control facilities and drive the transformation of the environmental governance mode from control-based end treatment to prevention-based clean production, thereby reducing pollution and enhancing environmental resilience. Attributed to advances in technological innovation and the concept of green development, the conventional production–circulation–consumption-discard linear mode is transformed into the green "production–circulation–consumption–discard–recycle" circular mode [50]. To sum up, DF plays a key role in promoting technological innovation and optimizing resource allocation, thereby possessing indirect impacts on GLCD [51].

**Hypothesis 2 (H2):** *DF can indirectly facilitate GLCD through technological innovation.*

This paper puts forward two research hypotheses for DF to promote GLCD, and establishes the mechanism of DF affecting GLCD (Figure 1).

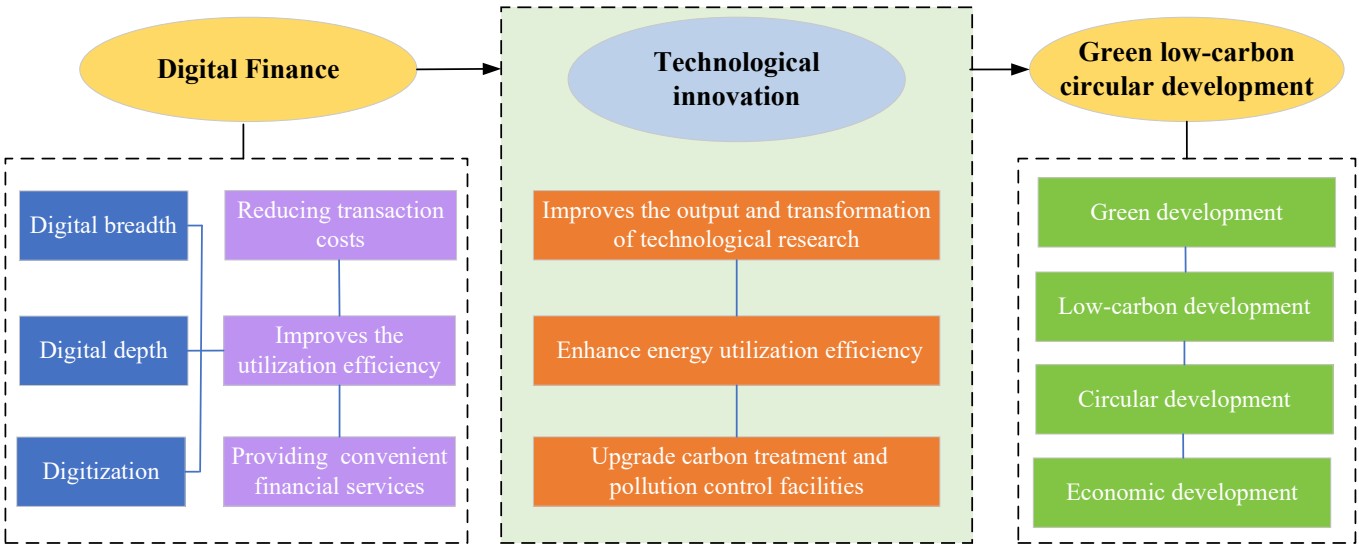

**Figure 1.** The mechanism of DF affecting GLCD.

## 3. Establishment of an Indicator System for GLCD

### 3.1. The Measure of GLCD

GLCD is essentially an integration of green development, low-carbon development, and circular development. Specifically, green development emphasizes rational utilization of natural resources and environmental protection, low-carbon development focuses on carbon emission reduction, and circular development aims at efficient resource utilization. Therefore, this research established a measurement system of GLCD in China (see Table 1) from four dimensions: green development, low-carbon development, circular development, and economic development, and calculates the comprehensive development level of GLCD in China by using spatiotemporal range entropy weight method.

**Table 1.** Comprehensive index system of GLCD.

| Primary Indicator | Secondary Indicator | Tertiary Indicator | Indexes |
|---|---|---|---|
| GLCD indicator system | Green development | Natural resources | Water resource per capita (m$^3$/capita) <br> The number of nature reserves <br> Forest coverage (%) |
| | | Ecological environment | Green coverage in urban areas (%) <br> The local green space area <br> Local forest area |
| | Low-carbon development | Carbon emission | Carbon emission/GDP (10,000 t/100 million yuan) <br> Carbon emission/population (10,000 t/10,000 residents) <br> Carbon emission/area |
| | | Carbon productivity | GDP/carbon emission (100 million yuan/10,000 t) <br> GDP of the secondary industry/carbon emission <br> GDP of the tertiary industry/carbon emission |

**Table 1.** *Cont.*

| Primary Indicator | Secondary Indicator | Tertiary Indicator | Indexes |
|---|---|---|---|
| GLCD indicator system | Circular development | Utilization capability | The comprehensive utilization rate of industrial solid waste (%)<br>Industrial water utilization rate (%)<br>Recycling water utilization rate in urban areas (%) |
| | | Processing capacity | Industrial waste gas treatment capacity (10,000 cm$^3$/h)<br>Sewage treatment rate in urban areas (%)<br>Hazard-free treatment rate of domestic waste (%) |
| | Economic development | Economic benefits | GDP per capita<br>The proportion of tertiary industry in GDP (%)<br>Total import/export volume/GDP (%) |
| | | Social benefits | The average salary of employees (yuan)<br>Public library collections per capita (copies)<br>The registered unemployment rate in urban areas (%) |

### 3.2. Method

At present, methods to determine indicator weights include subjective weighting methods, objective weighting methods, and integrated weighting methods. In this paper, the GLCD level of 31 provinces in China from 2008 to 2019 is measured by using the spatiotemporal range entropy weight method. Conventional entropy weight methods can only determine indicator weights at a specific moment, while the spatiotemporal range entropy weight method can determine indicator weights in both temporal and spatial dimensions, thereby updating indicator weights dynamically.

Assume that a multi-level evaluation system consists of $n$ evaluation indicators of $m$ evaluation objects in $k$ periods. The indicator system is $X_i$ ($i = 1, 2, \ldots, n$), and its indicator in period $t$ is $X_{ijt}$ ($j = 1, 2, \ldots, m$ and $t = 1, 2, \ldots, k$).

(1) Data standardization: let $x_{ijt}$ be $X'_{ijt}$, a dimensionless decision matrix can be obtained: If $x_i$ is a positive indicator,

$$x_{ijt}' = \left[x_i - \min(x_{ijt})\right] / \left[\max(x_{ijt}) - \min(x_{ijt})\right] \tag{1}$$

If $x_i$ is a negative indicator,

$$x_{ijt}' = \left[max(x_{ijt}) - x_{ijt}\right] / \left[\max(x_{ijt}) - \min(x_{ijt})\right] \tag{2}$$

(2) Calculation of weights based on information entropy:

$$e_i = -ln(km)^{-1}\sum_j \sum_t p_{ijt} ln(p_{ijt}) \tag{3}$$

where

$$p_{ijt} = x_{ijt}' / \sum_j \sum_t x_{ijt}' \tag{4}$$

(3) The weight of $X_i$ can be calculated by:

$$W_i = (1 - e_i) / \left(\sum_i 1 - e_i\right) \tag{5}$$

(4) Weighted summation of various indicators:

$$Z_{it} = \sum W_i \times x_{ijt} \tag{6}$$

## 4. Research Design

*4.1. Model Establishment*

### 4.1.1. Benchmark Regression Model

To capture and validate Hypothesis 1, this paper developed a panel benchmark model. The panel benchmark model is expressed as follows:

$$GLCD_{i,t} = \alpha + \beta_1 DF_{i,t} + \beta_2 Edu_{i,t} + \beta_3 Fdi_{i,t} + \beta_4 Gov_{i,t} + \beta_5 Pop_{i,t} + u_i + \varepsilon_{i,t} \quad (7)$$

where $i$ denotes the city, $t$ denotes the year, and GLCD denotes the green low-carbon circular development, which indicates the high-quality economic development level; *DF* is the digital finance index, which reflects the regional digital economy development level; $Edu_{i,t}$ represents the education level; $Fdi_{i,t}$ represents the direct foreign investment level; $Gov_{i,t}$ represents the government expenditure level; $Pop_{i,t}$ represents the population level; $u_i$ represents the individual fixed effect; $\varepsilon_{i,t}$ represents the error term.

### 4.1.2. Mediating Effect Model

To clarify the transmission mechanism of DF affecting GLCD, a mediating effect model is constructed with technological innovation as follows:

$$TEC_{i,t} = \alpha + \beta_1 DF_{i,t} + \beta_2 Edu_{i,t} + \beta_3 Fdi_{i,t} + \beta_4 Gov_{i,t} + \beta_5 Pop_{i,t} + u_i + \varepsilon_{i,t} \quad (8)$$

$$GLCD_{i,t} = \alpha + \lambda TEC_{i,t} + \beta_1 DF_{i,t} + \beta_2 Edu_{i,t} + \beta_3 Fdi_{i,t} + \beta_4 Gov_{i,t} + \beta_5 Pop_{i,t} + u_i + \varepsilon_{i,t} \quad (9)$$

where $i$ denotes the city, $t$ denotes the year, and *TEC* is the mediating variable, representing technological innovation level; $u_i$ represents the individual fixed effect; $\varepsilon_{i,t}$ represents the error term.

### 4.1.3. Threshold Effect Model

The impact of DF and technological innovation on GLCD was investigated by using a panel benchmark regression model and a mediating effect model. Due to the "online effect", the impacts of DF on GLCD may be nonlinear. In this section, the linearity/nonlinearity of such impacts is further investigated and verified by using the following threshold regression model:

$$\begin{aligned} GLCD_{i,t} \quad &= \alpha + \beta_1 DF_{i,t} I(DF_{i,t} \le \gamma_1) + \beta_2 DF_{i,t} I(\gamma_1 < DF_{i,t} \le \gamma_2) + \\ &\beta_n DF_{i,t} I(\gamma_{n-1} < DF_{i,t} \le \gamma_2) + \beta_n + DF_{i,t} I(DF_{i,t} > \gamma_n) + \\ &\phi X_{i,t} + \varepsilon_{i,t} \end{aligned} \quad (10)$$

To further explore the nonlinearity of the impacts of DF on green low-carbon circular development, a nonlinear effect model is developed with innovation input and innovation output as the threshold variables (*TEC*):

$$\begin{aligned} GLCD_{i,t} \quad &= \alpha + \beta_1 DF_{i,t} I(TEC_{i,t} \le \gamma_1) + \beta_2 DF_{i,t} I(\gamma_1 < TEC_{i,t} \le \gamma_2) + \\ &\beta_n DF_{i,t} I(\gamma_{n-1} < TEC_{i,t} \le \gamma_2) + \beta_n + DF_{i,t} I(TEC_{i,t} > \gamma_n) + \\ &\phi X_{i,t} + \varepsilon_{i,t} \end{aligned} \quad (11)$$

where $\gamma_1, \gamma_2, \ldots, \gamma_n$ are the thresholds to be estimated; $I(\cdot)$ is the indicator function, and its value is based on the expression in parentheses; *X* represents the control variable, which is the same that in Equation (1).

### 4.1.4. Machine Learning Model

As the impact of DF on GLCD is affected by various elements, multicollinearity and DOF degradation can be alleviated by using the nonlinear random forest model and the nonlinear CatBoost model in machine learning. The contributions of variables are calculated

by establishing the random forest model and the CatBoost model, respectively, to clarify the effects of different variables on GLCD.

In random forest regression tree models, splitting nodes are selected by taking the least mean square error as the optimization criteria, and the variables are sorted according to their contributions to the reduction in the residual sum of square:

$$\min_{j,m}\left[\min\sum\nolimits_{Yi\in R_1(j,m)}(Yi-\overline{Y_{R1}})^2+\min\sum\nolimits_{Yi\in R_2(j,m)}(Yi-\overline{Y_{R2}})^2\right] \tag{12}$$

where $\overline{Y_{R1}}$ and $\overline{Y_{R2}}$ are the determined mean of each spurious copy of output variables; $j$ and $m$ refer to the splitting nodes of variables and threshold, respectively.

In the CatBoost model, more effective strategies are adopted to reduce over-fitting, and the entire dataset is used for training to effectively utilize the data information:

$$x_{i,k}=\frac{\sum_{j=1}^{p-1}\left[x_{\sigma_j,k}=x_{\sigma_p,k}\right]\times Y_j+\alpha\times p}{\sum_{j=1}^{p-1}\left[x_{\sigma_j,k}=x_{\sigma_p,k}\right]+\alpha} \tag{13}$$

where $\sigma_j$ is the $j$th data, $x_{i,k}$ is the $k$th discrete feature of the $i$th data in the training set, $a$ is a priori weight, and $p$ is a priori distribution term (Table 2).

**Table 2.** The summary of all methods.

| Method | Explanation |
| --- | --- |
| Benchmark regression model | To clarify the effect of DF on GLCD |
| The mediation effect model | To clarify the mechanism of DF affecting GLCD |
| Threshold regression | To Prove whether there is a nonlinear characteristic in the effect of DF on GLCD |
| Machine learning model | To examine the relative importance of DF and the factors affecting GLCD |

*4.2. Variable Selection*

4.2.1. Explained Variables

In this study, a multi-dimensional evaluation system consisting of four dimensions (economic development, green development, low-carbon development, and circular development) is constructed, with the GLCD level measured by the spatio-temporal range entropy weight method as explained variable.

4.2.2. Key Explanatory Variable

In this study, the regional DF development level is investigated from three dimensions: breadth (*Bre*), depth (*Dep*), and digitization (*Dig*), with the total index of digital inclusive finance (*DF*) of China as the proxy variable of regional DF development level.

4.2.3. Mediating Variables

In this study, regional technological innovation is evaluated based on the proportion of science and technology expenditure in financial expenditure (*Sci*) and the number of patent applications per capita (*Pat*). This is consistent with the investigation of the influences of DF on GLCD through technological innovation.

4.2.4. Control Variables

To clarify the effect of DF on GLCD, the control variables affecting GLCD should also be set: (1) education level (*Edu*), which is evaluated based on the proportion of education expenditure in GDP; (2) direct foreign investment (*Fdi*), which is evaluated based on the proportion of direct foreign investment in GDP; (3) population (*Pop*), which is evaluated based on the permanent residents in this region; (4) government expenditure (*Gov*), which is evaluated based on the proportion of government expenditure in GDP.

### 4.3. Data Sources and Descriptive Statistics

In this study, the effect of DF on GLCD is investigated in terms of 31 provinces in China. The variables come from the Statistical Yearbook of China, the Energy Statistical Yearbook of China, and the Environment Statistical Yearbook of China from 2012 to 2020. The digital inclusive finance index comes from the Digital Inclusive Finance Indicator System and Index Compilation released by the Institute of Internet Finance, Peking University. The technological innovation data are collected from the website of the World Intellectual Property Organization (WIPO); missing data are determined by the interpolation method. The descriptive statistical results of the main variables are listed in Table 3.

**Table 3.** Descriptive statistics for all variables.

| Variables | Symbol | Calculation Method |
|---|---|---|
| Green low-carbon circular development | GLC | Spatio-temporal range entropy weight method |
| | DF | The digital finance level |
| Digital finance | Bre | Breadth |
| | Dep | Depth |
| | Dig | Digitization |
| technological innovation | Sci | The proportion of science and technology expenditure in financial expenditure (%) |
| | Pat | The number of patent applications per capita (/10,000 residents) |
| education level | Edu | The proportion of education expenditure in GDP (%) |
| direct foreign investment | Fdi | The proportion of direct foreign investment in GDP (%) |
| population | Pop | Permanent population (10,000 residents) |
| government expenditure | gov | The proportion of financial expenditure in GDP (%) |

## 5. Empirical Analysis

### 5.1. Benchmark Analysis

Table 4 presents the benchmark regression results. Model 1 corresponds to the estimated result with considering control variables. The results show that DF significantly promoted an improvement of GLCD. Promoting digital finance was conductive to developing green low-carbon circular development. The regression results (Table 4, Model 1) showed that for every 1% increase in DF, GLCD increased by 0.014%, The cause of this phenomenon is that "Data + Algorithm + Computing Power" can break temporal and spatial limitations to achieve rapid flow of various resource elements, thereby realizing effective docking and precise matching. It may continuously induce revolutions in the industry and facilitate the green transformation of energy-intensive industries. Model 2–Model 4 denote the dimension-reduction regression results of the effect of breadth (Bre), depth (Dep), digitization (Dig) on GLCD. The regression results (Table 4, Model 2–Model 4) showed that for every 1% increase in Bre, Dep, and Dig, GLCD increased by 0.011%, 0.011%, and 0.013%, respectively. This can be attributed to the fact that digital techniques such as artificial intelligence, big data, cloud computing, and blockchain significantly enhance labor productivity and decision-making efficiency, improving the efficiency of environmental management, ultimately leading to a reduction in energy consumption, thereby promoting GLCD. This proves the validity of Hypothesis H1.

Furthermore, the analysis of the control variables yields insightful results. The coefficient of Edu exhibits a significant positive relationship, indicating that higher levels of education level contribute to a promotion in GLCD. In contrast, the coefficient of Fdi demonstrates a negative association, confirming that foreign direct investment growth leads to a decrease in GLCD. The expansion of the foreign direct investment stimulates production and daily activities, resulting in higher energy demands and pollution emissions, inhibiting GLCD levels. However, the coefficient of Pop and Gov fail to pass the significance test, indicating that the effect of population and government expenditure on GLCD has not been effectively demonstrated. Despite the recent intensification of government expenditure on green development in China, their actual projects need improvement.

**Table 4.** The benchmark regression results.

| Variables | Baseline Regression | Dimension-Reduction Regression | | |
|---|---|---|---|---|
| | Model 1 | Model 2 | Model 3 | Model 4 |
| DF | 0.014 *** | | | |
| | (0.002) | | | |
| Bre | | 0.011 *** | | |
| | | (0.001) | | |
| Dep | | | 0.011 *** | |
| | | | (0.002) | |
| Dig | | | | 0.013 *** |
| | | | | (0.002) |
| Edu | 0.203 * | 0.188 | 0.178 | 0.210 * |
| | (0.116) | (0.116) | (0.119) | (0.114) |
| Fdi | −0.037 *** | −0.035 *** | −0.040 *** | −0.037 *** |
| | (0.010) | (0.010) | (0.010) | (0.010) |
| Pop | 0.068 | 0.086 | 0.142 * | 0.091 |
| | (0.077) | (0.076) | (0.078) | (0.070) |
| Gov | −0.002 | 0.007 | 0.027 | 0.004 |
| | (0.060) | (0.059) | (0.061) | (0.057) |
| cons | −0.265 | −0.390 | −0.849 | −0.453 |
| | (0.622) | (0.615) | (0.628) | (0.565) |
| N | 279 | 279 | 279 | 279 |
| $R^2$ | 0.223 | 0.219 | 0.181 | 0.256 |

Statistical significance at: * $p < 0.01$, *** $p < 0.01$. The t-values are in parentheses. The Hausman test is conducted to assess the appropriateness of random or fixed effects models. Rejecting Ho indicates that the fixed effect is valid.

### 5.2. Mechanism Analysis

The benchmark regression results illustrate that DF contributed toward improving GLCD. However, we have yet to determine whether the interaction mechanism between DF and GLCD conforms to the above theoretical analysis. Therefore, we further tested whether DF promotes GLCD improvement through certain mechanisms, such as technological innovation effects.

Table 5 shows the regression results of the mechanism test. Model 1 and Model 3 demonstrate that the mechanism test of science and technology expenditure in financial expenditure (Sci) and the number of patent applications per capita (Pat). Model 2 and Model 4 demonstrate the results of the mediating effects with the mediating variables. The coefficient of DF is statistically positive (passing the significance test at the 1% level) when Sci and Pat are explanatory variables, demonstrating that DF has a mediating influence on promoting GLCD. Therefore, the transmission mechanism that DF affects GLCD through technological invention transformation is valid.

**Table 5.** Analysis of mediating effects.

| Variables | Sci | GLCD | Pat | GLCD |
|---|---|---|---|---|
| | Model 1 | Model 2 | Model 3 | Model 4 |
| Sci | | 0.959 *** | | |
| | | (0.271) | | |
| Pat | | | | 0.001 *** |
| | | | | (0.000) |
| DF | 0.001 *** | 0.012 *** | 1.235 *** | 0.012 *** |
| | (0.000) | (0.002) | (0.380) | (0.002) |
| Edu | −0.006 | 0.209 * | −31.020 * | 0.249 ** |
| | (0.027) | (0.113) | (17.240) | (0.114) |
| Fdi | 0.002 | −0.040 *** | 8.615 *** | −0.050 *** |
| | (0.002) | (0.010) | (1.547) | (0.010) |
| Pop | 0.050 *** | 0.021 | 28.010 ** | 0.027 |
| | (0.018) | (0.076) | (11.500) | (0.076) |
| Gov | −0.055 *** | 0.050 | −13.690 | 0.017 |
| | (0.014) | (0.061) | (8.920) | (0.059) |
| cons | −0.379 *** | 0.098 | −221.700 ** | 0.0626 |
| | (0.144) | (0.616) | (92.300) | (0.615) |
| N | 279 | 279 | 279 | 279 |
| $R^2$ | 0.193 | 0.262 | 0.413 | 0.261 |

Statistical significance at: * $p < 0.01$,** $p < 0.05$, *** $p < 0.01$. The t-values are in parentheses.

### 5.3. Threshold Regression

We used the benchmark regression model and mediation effect model to verify the effect of DF on GLCD through technological innovation in previous sections. In this section we use the threshold effect model to test the existence and number of panel thresholds (see Table 6). It was found that only the single threshold passed the significance test at 5% in terms of DF, with a threshold value of 5.38. This suggested that there was a single threshold for the impact of the DF on GLCD. In terms of Sci, the single threshold passed the test at the significance level of 5% with a value of 0.033, while the significance test was not passed for the double threshold. In terms of Pat, the significance test was not passed for the single and double thresholds. This showed that there was no threshold based on the increase in patents. To further investigate the threshold effect of DF on GLCD, the single threshold models were estimated based on the threshold value test, with the results reported in Table 7.

**Table 6.** Verification of threshold effects.

| Threshold Variables | Threshold Numbers | Threshold Value | Critical Value | | | |
|---|---|---|---|---|---|---|
| | | | **F** | **1%** | **5%** | **10%** |
| *DF* | Single threshold | 5.380 ** | 28.65 | 31.684 | 21.882 | 18.675 |
| | Double threshold | 4.610 | 8.62 | 22.975 | 16.118 | 13.459 |
| *Sci* | Single threshold | 0.033 ** | 28.32 | 31.849 | 25.960 | 22.517 |
| | Double threshold | 0.056 | 4.65 | 25.842 | 22.133 | 18.155 |
| *Pat* | Single threshold | 1.138 | 14.48 | 40.778 | 28.654 | 24.545 |

Statistical significance at: ** $p < 0.05$, The t-values are in parentheses.

**Table 7.** The results of threshold regression.

| Variables | DF | | Sci | |
|---|---|---|---|---|
| | **Coef** | **T** | **Coef** | **T** |
| $DF_1$ (DF ≤ 5.38) | 0.010 *** | 2.39 | | |
| $DF_2$ (DF > 5.38) | 0.013 *** | 3.66 | | |
| $DF_1$ (Sci ≤ 0.033) | | | 0.013 *** | 5.33 |
| $DF_2$ (Sci > 0.033) | | | 0.019 *** | 7.24 |
| Control variables | Yes | Yes | Yes | Yes |
| $R^2$ | 0.308 | | 0.306 | |
| N | 279 | | 279 | |
| F | 15.39 *** | | 15.23 *** | |

Statistical significance at: *** $p < 0.01$. The t-values are in parentheses.

Table 7 summarizes the regression results of threshold models. As indicated by the table, the impacts of DF on GLCD are positively nonlinear. In the first case, DF is employed as the threshold variable for threshold estimation. If the DF development level is lower than the threshold (5.38), the estimation coefficient of DF on GLCD is 0.010; if the DF development level is higher than the threshold (5.38), the estimation coefficient of DF on GLCD is 0.013. The results indicate that the positive impacts of DF on GLCD depend on the DF system, and such impacts are more significant in the late stage. In the second case, the innovation input is taken as a threshold variable for threshold estimation. If Sci is lower than the threshold (0.033), the positive impacts of DF on GLCD are low; if the SCi is higher than the threshold (0.033), the positive impacts of DF on GLCD are greatly enhanced. When the innovation level varies, the impacts of DF on GLCD change with the innovation input; as the innovation input increases, the impacts of DF on GLCD exhibit significant positive nonlinear features with increasing "marginal effect".

## 6. Discussion

### 6.1. Interpretation of Findings

The promotion of DF significantly improved GLCD during the study period. Technological innovation effect promoted GLCD, which supports Hypothesis 1 (i.e., DF can promote GLCD). Therefore, exploring the impact of DF on GLCD is an important measure and path choice to achieve "carbon peak, carbon neutralization". At the same time, it should be noted that the current theoretical research on the impact of DF on GLCD is still in its infancy, and its mechanism research has only obtained some preliminary results. The green optimization effect of DF will be the key content of theoretical research to achieve GLCD in the future. In addition, in view of the heterogeneity of different regions in China, in-depth exploration of the regional suitability and applicability of DF affecting GLCD will be the core proposition that must be tackled when green theory is applied to practice. Pilot policies can make important contributions to GLCD in digital inclusive financial reform pilot zone and digital financial service platform. The establishment of the national digital inclusive financial reform pilot zone has accumulated experience in the exploration of financial support for the development of economic civilization and ecological civilization. The digital inclusive financial reform pilot zone has achieved positive results, and through accelerating the innovation of digital financial inclusion and other pilot means, guides funds to invest in green fields. More importantly, the mode of financial production and service must also be changed from high carbon to low carbon or even zero carbon.

### 6.2. Machine Learning Model of Nonlinear Effects

The importance of explanatory variables, mediating variables, and control variables that affect GLCD can be identified using the random forest model and the CatBoost model, and these variables are sorted according to their impacts on GLCD. According to Figures 2 and 3, GLCD has a significantly positive correlation with DF, with a contribution rate of 2.2% and 9.8%, respectively, indicating that GLCD can benefit from DF. Meanwhile, the other six characteristic variables have significantly positive impacts on GLCD, which is consistent with the expectation. Herein, the proportion of technology expenditure and population exhibit significantly higher contribution rates than other economic factors, which indicates that technology and talent are determining factors in economic development. Technology is the primary factor of production, and talent is the primary resource, and they can promote GLCD.

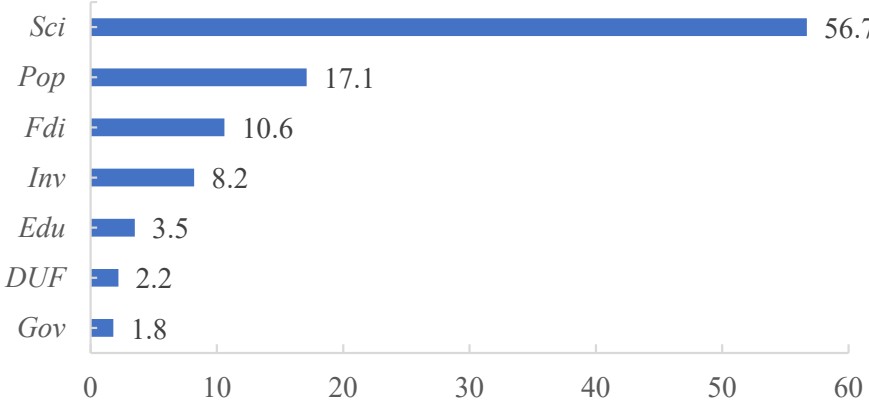

**Figure 2.** The significance of different variables (by the random forest model).

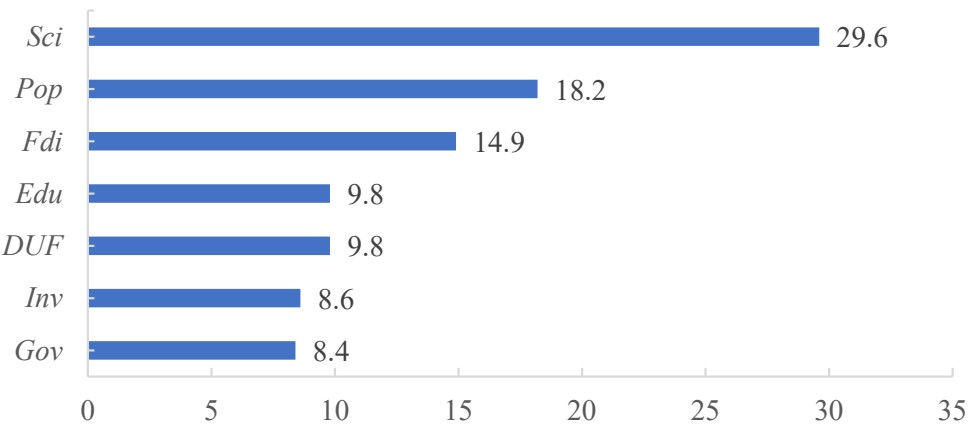

**Figure 3.** The significance of different variables (by the CatBoost model).

*6.3. Heterogeneity Analysis*

The impacts of DF on GLCD and the transmission mechanism, as well as the threshold effect, have been discussed in previous sections. However, regional differences, which lead to varying impacts of DF on GLCD, are not considered. Table 8 shows the impacts of DF on GLCD in different regions and basins. First, 31 provinces are divided into eastern, central, and western China, as shown in Model 1–Model 3, respectively. Both "Development of the Yangtze River Economic Belt" and "Ecological Conservation and High-Quality Development of the Yellow River Basin" have become key national strategies and benchmarks for the governance of large rivers in China. To clarify such differences from the perspective of national strategy, the Yangtze River Economic Belt and the Yellow River Basin are taken for regression analysis, where the former includes Shanghai, Jiangsu, Zhejiang, Anhui, Jiangxi, Hubei, Hunan, Chongqing, Sichuan, Yunnan, and Guizhou, and the latter includes Qinghai, Sichuan, Gansu, Ningxia, Inner Mongolia, Shaanxi, Shanxi, Henan, and Shandong, as shown in Model 4 and Model 5.

**Table 8.** The heterogeneity regression results.

| Variables | East | Middle | West | Yangtze River Belt | Yellow River Basin |
|---|---|---|---|---|---|
| | Model 1 | Model 2 | Model 3 | Model 4 | Model 5 |
| DUF | 0.007 | 0.017 *** | 0.021 *** | 0.023 *** | 0.017 *** |
| | (0.005) | (0.004) | (0.003) | (0.005) | (0.005) |
| Edu | 0.121 | 0.114 | 0.236 | −0.202 | 0.306 |
| | (0.224) | (0.185) | (0.180) | (0.218) | (0.222) |
| Fdi | −0.027 ** | 0.040 | −0.082 *** | −0.065 ** | −0.085 ** |
| | (0.013) | (0.050) | (0.028) | (0.029) | (0.038) |
| Pop | 0.162 | 0.272 | −0.210 * | −0.109 | −0.227 |
| | (0.133) | (0.198) | (0.110) | (0.222) | (0.311) |
| Gov | −0.156 | −0.369 *** | 0.276 *** | 0.049 | 0.056 |
| | (0.144) | (0.118) | (0.075) | (0.142) | (0.123) |
| cons | −0.915 | −1.970 | 1.695 ** | 1.275 | 2.010 |
| | 1.093 | (1.663) | (0.821) | (1.889) | (2.488) |
| N | 99 | 90 | 90 | 99 | 81 |
| $R^2$ | 0.124 | 0.401 | 0.476 | 0.348 | 0.263 |

Statistical significance at: * $p < 0.01$, ** $p < 0.05$, *** $p < 0.01$. The t-values are in parentheses.

According to the sub-region heterogeneity regression results obtained by Model 1–Model 3 shown in Table 8, the impact of DF on GLCD are maximized in western China. This can be attributed to the fact that central and western China have abundant resources and energy-intensive industries. DF can relieve this situation by significantly promoting digitalization and low-carbon transformation of traditional industries in western China. Specifically, energy digitalization will accelerate the growth of energy conservation and

environmentally-friendly industries, and continuously facilitate the low-carbon development of all industries.

According to the sub-region heterogeneity regression results obtained by Model 4 and Model 5, the impacts of DF on GLCD are significantly positive in both the Yangtze River Economic Belt and the Yellow River Basin. This is because DF can enhance the green technological innovation capability, optimize industrial structure and spatial layout in the region, and improve the utilization efficiency of energy and resources, thereby facilitating GLCD.

*6.4. Robustness Testing*

To demonstrate the reliability of the empirical results obtained, robustness testing is conducted in three aspects: (1) Beijing, Shanghai, Tianjin, and Chongqing are excluded due to their unique economic development level and economic policies. As indicated by Model 1 in Table 9, the regression coefficient of DF on GLCD remains significantly positive (verified by a 1% significance level test), which is consistent with the results of baseline regression analysis. (2) Instrumental variable method: to mitigate endogeneity, estimation is conducted by using the two-stage least squares (2SLS) method with Internet popularity as the instrumental variable. As indicated by Model 2 in Table 9, the Internet popularity coefficient is significantly positive, which agrees with the conclusions of previous studies. (3) Change variation method: robustness testing is carried out by introducing the lagging phase of GLCD into the model and using mixed OLS and dynamic panel system GMM models, as indicated by Model 3 and Model 4 in Table 9. Model 3 refers to the mixed OLS regression, and the coefficient is significantly positive. As illustrated in Model 4, the regression coefficient of the previous phase of GLCD level on the current phase of GLCD level is significantly positive. That is, the process of GLCD evolves dynamically, and the regression coefficient of DF on GLCD levels remains significantly positive, indicating the high stability of the regression results.

**Table 9.** Robustness testing.

| Variables | Model 1 | Model 2 | Model 3 | Model 4 |
|---|---|---|---|---|
| L.GLC | | | | 0.955 *** |
| | | | | (0.037) |
| DUF | 0.015 *** | 0.046 *** | 0.014 *** | 0.022 * |
| | (0.002) | (0.013) | (0.002) | (0.013) |
| Edu | 0.221 * | 0.847 *** | 0.328 ** | −0.056 |
| | (0.125) | (0.198) | (0.158) | (0.520) |
| Fdi | −0.049 *** | 0.116 *** | −0.008 | 0.002 |
| | (0.012) | (0.015) | (0.018) | (0.004) |
| Pop | 0.065 | 0.030 ** | 0.009 | 0.005 |
| | (0.089) | (0.009) | (0.020) | (0.003) |
| Gov | −0.000 | 0.175 | 0.012 | 0.026 |
| | (0.065) | (0.031) | (0.095) | (0.010) |
| cons | −0.264 | −0.337 ** | 0.179 | −0.143 |
| | (0.723) | (0.069) | (0.179) | (0.075) |
| AR(2) | | | | 0.559 |
| Hansen | | | | 0.459 |
| $R^2$ | 0.237 | 0.309 | 0.197 | |

Statistical significance at: * $p < 0.01$, ** $p < 0.05$, *** $p < 0.01$. The t-values are in parentheses.

## 7. Conclusions

Combined with theoretical analysis, this study evaluated the effect of DF on GLCD, analyzed the mechanism of action, and examined the heterogeneity effects between the two. The conclusions are summarized below.

According to the benchmark analysis results, DF significantly promoted GLCD. A series of robustness and endogeneity analyses also supported this conclusion. Specifically, the technological innovation effect contributed to the promotion mechanism of DF. DF

has an increasing "marginal effect" on GLCD with its nonlinear characteristics. Moreover, a machine learning model is used to identify the importance of explanatory variables, mediating variables, and control variables that affect GLCD. We found that technological innovation contributes most to the impact of GLCD. Additionally, the promotion effect in China's central and western regions was greater than that in the eastern region. This can be attributed to the fact that central and western regions have abundant resources and energy-intensive industries, and energy digitalization will accelerate the growth of energy conservation and environmentally-friendly industries, and continuously facilitate green low-carbon development of all industries.

The research outcome of this paper provides consolidation of the evidence on the positive effects of DF on GLCD by the empirical research from the perspectives of the ecological economy and digital economy. With this study, we propose the following suggestions to make use of DF to facilitate GLCD.

Firstly, we suggest accelerating green transformation in key industries and fields to promote clean production and develop environmentally-friendly industries. This would further help to achieve low-carbon, safe, and efficient energy utilization of treatment sludge of domestic waste and sewage. For this reason, we should enhance the recycling of urban recycled water, and integrate the concept of green development into industry and daily life, to reduce carbon emission.

Secondly, we suggest that DF infrastructure should be constructed to facilitate the digital transformation of various industries, and the empowering channels of DF shall be expanded. Additionally, we should stimulate the innovation capability of DF and improve digital productivity, as well as strengthen the positive impacts of DF on GLCD.

Thirdly, based on regional differences in development levels, we suggest encouraging cooperation between regions, strengthening cooperation and mutual assistance, and market integration of different regions and basins with different resource endowments, and to improve the infrastructure of DF, in order to further unleash the spillover dividends of DF for green development.

Last but the least, technical cooperation and transactions can be achieved with the assistance of user-friendly and efficient digital platforms, which would help to promote the synergistic development of developed and underdeveloped regions and exploit the positive impact of DF on GLCD.

**Author Contributions:** Conceptualization, X.Z. and X.A.; Data curation, X.Z.; Funding acquisition, G.Z.; Methodology, X.Z.; Resources, X.Z. and X.A.; Software, X.Z.; Supervision, X.W.; Writing—original draft, X.Z.; Writing—review and editing, X.A., X.W. and J.Z. All authors have read and agreed to the published version of the manuscript.

**Funding:** This research was supported by the Innovative Research Group Project of the National Natural Science Foundation of China (NSFC) (Grant no.42121001), the National Natural Science Foundation of China (NSFC) (Grant Nos.72364027 and 72164030), the Natural Science Foundation of Inner Mongolia (Grant No.2023QN07008), and the Inner Mongolia Foundation of Philosophy and Social Sciences (Grant No.2022NDB134). All authors approved the version of the manuscript to be published.

**Data Availability Statement:** Data is available upon request to any of the authors.

**Conflicts of Interest:** The authors declare no conflict of interest.

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
