# Peer review of "A Study on the Effects of Digital Finance on Green Low-Carbon Circular Development Based on Machine Learning Models"

_mathematics, doi:10.3390/math11183903_

Round 1

Reviewer 1 Report

First, the authors need to formulate the problem they are solving in their research.

Secondly, to specify the purpose of the study in accordance with the problem and hypotheses of the study.

First, the authors need to formulate the problem they are solving in their research.

Secondly, to specify the purpose of the study in accordance with the problem and hypotheses of the study.

Reviewer 2 Report

The authors investigated "...the effects of digital finance on green and low-carbon cyclical development and the mechanism. First, the level of green and low-carbon cyclical development in China is estimated from multiple aspects based on the panel data of 31 provinces using the spatio-temporal range entropy weighting method. Then, the working mechanism of digital finance in green and low-carbon cyclical development is revealed by conducting empirical tests based on panel regression models, mediating effect models and threshold models...". The paper has great potential, however the authors should improve the following items:

1) In the abstract the authors explain well how they will carry out the research, but they should improve the results/findings as well as their practical implications;

2) The introduction should present the gap in the literature, clearly explain the major objective of the study, as well as the contribution of this study to the existing literature;

3) The Literature Review is robust and well done, the authors can improve by making summaries after raising the research hypotheses;

4) In the methodology, the authors should make a summary table of the various techniques to be used, for a better interpretation and explain why this methodological?

5) In the results the authors focus too much on the models and should better explain the findings;

6) In the conclusion the authors should explain the findings and should make a new point explaining the practical implications of these findings.

Authors should make minor revisions in English. 

Reviewer 3 Report

I thoroughly read the paper entitled "A study on the effects of digital finance on green and low carbon cyclic development based on machine learning models" in which I feel that the paper has some potential to publish in this journal. But before to publish, the paper should majorly revised.

1. Please improve the Introduction part, in term of theory, novelty, objectives, and literature. Drastically revise the Introduction section.

2. I suggest that for theory and mechanism authors should draw graph/figure then every reader can understand the idea of the paper.

3. Why the authors utilized machine learning model instead of other models. In best of my knowledge machine learning model is good fit for secondary data but here the authors use Panel data.

4. Discussion part is weak please extend this part.

Please check the English language of the paper

4. 

Please check the English language of the paper

Round 2

Reviewer 1 Report

Paper can be accepted in present form

Paper can be accepted in present form

Reviewer 3 Report

After careful reading I accept the paper in the current form.